# ChatGPT to Replace Crowdsourcing of Paraphrases for Intent Classification: Higher Diversity and Comparable Model Robustness

**Jan Cegin**♠†**, Jakub Simko**†**, Peter Brusilovsky**‡

♠ Faculty of Information Technology, Brno University of Technology, Brno, Czechia
† Kempelen Institute of Intelligent Technologies, Bratislava, Slovakia
{jan.cegin, jakub.simko}@kinit.sk
‡ University of Pittsburgh, Pittsburgh, USA
peterb@pitt.edu

## Abstract

The emergence of generative large language models (LLMs) raises the question: what will be its impact on crowdsourcing? Traditionally, crowdsourcing has been used for acquiring solutions to a wide variety of human-intelligence tasks, including ones involving text generation, modification or evaluation. For some of these tasks, models like ChatGPT can potentially substitute human workers. In this study, we investigate whether this is the case for the task of paraphrase generation for intent classification. We apply data collection methodology of an existing crowdsourcing study (similar scale, prompts and seed data) using ChatGPT and Falcon-40B. We show that ChatGPT-created paraphrases are more diverse and lead to at least as robust models.

## 1 Introduction

Crowdsourcing has been an established practice for collecting training or validation examples for NLP tasks, including the *intent classification* (i.e. determining the purpose behind a phrase or sentence). When crowdsourcing intent examples, workers typically create new phrases for some scenario (Wang et al., 2012; Larson et al., 2020). However, a data augmentation approach can also be used: by workers *paraphrasing* existing sentences with known intents. The aim is to increase the *data diversity* and subsequent *model robustness*. In particular, diversity can increase considerably with the use of taboo words that force workers to be more creative (Larson et al., 2020). Without data diversity, models can lose their ability to generalize well (Larson et al., 2020; Joshi and He, 2022; Wang et al., 2022).

Unfortunately, crowdsourcing has several downsides: 1) the workforce is costly, 2) output quality is difficult to achieve (which translates to further costs), and 3) there are overheads related to the design and organization of the process.

The advent of generative large language models (LLMs) and ChatGPT in particular opens up the possibility of *replacement of crowd workers by AI*. LLMs have been already investigated for a variety of NLP tasks: translation (Jiao et al., 2023), question answering, sentiment analysis, or summarization (Qin et al., 2023), displaying strong performance against fine-tuned models. Indeed, some crowd workers have already started to exploit LLMs to replace themselves (Veselovsky et al., 2023).

With this paper, *we investigate whether ChatGPT can replace crowd workers in paraphrase generation for intent classification dataset creation.* We answer the following research questions:

1. *RQ1: Can ChatGPT generate valid solutions to a paraphrasing task, given similar instructions as crowd workers?*

2. *RQ2: How do ChatGPT and crowd solutions compare in terms of their lexical and syntactical diversity?*

3. *RQ3: How do ChatGPT and crowd solutions compare in terms of robustness of intent classification models trained on such data?*

To answer these questions, we have conducted a quasi-replication of an existing study (Larson et al., 2020), where paraphrases were crowdsourced to augment intent classification data (using also taboo words technique to induce more example diversity). In our own study, we followed the same protocol (seeds, taboo words, scale), but replaced the crowd workers with ChatGPT (for approximately 1:600 lesser price) as the most widely used LLM and Falcon-40B (Almazrouei et al., 2023) as one of the best performing open source LLM at the time of writing this paper. This allowed us to directly compare properties of crowd- and LLM-generated data, with following findings: 1) ChatGPT is highly

reliable in generating valid paraphrases, 2) Falcon-40B struggled in generating valid and unique paraphrases, 3) ChatGPT outputs lexically and syntactically more diverse data than human workers, and 4) models trained on ChatGPT data have comparable robustness to those trained on crowd-generated data.

## 2 Related work: Collecting paraphrases and using ChatGPT

Crowdsourcing of paraphrases is used for creating new datasets for various NLP tasks (Larson et al., 2019a, 2020). In paraphrase crowdsourcing, the process typically entails showing of an initial seed sentence to the worker, who is then asked to paraphrase the seed to new variations (Ravichander et al., 2017). As training sample diversity is correlated with robustness of downstream models (Larson et al., 2020; Joshi and He, 2022; Wang et al., 2022), various diversity incentives are used to encourage crowd workers to create more diverse paraphrases. In some approaches, workers are hinted with word suggestions (Yaghoub-Zadeh-Fard et al., 2019; Rhys Cox et al., 2021). Syntactic structures can also be suggested (Ramírez et al., 2022). Another approach is *chaining*, where paraphrases from previous workers are used as seeds (Rhys Cox et al., 2021). Yet another technique is the use of *taboo* words, where users are explicitly told to avoid certain phrases. Previous research has shown, that taboo words lead to more diverse paraphrases and more robust models (Larson et al., 2020; Ramírez et al., 2022). Yet, despite all the advances, crowdsourcing remains expensive for dataset building.

Seq2seq models LLMs have already found their use for paraphrase generation from existing corpora (GPT-2 (Radford et al., 2019), BART (Lewis et al., 2020)). LLMs can also be used to paraphrase using style transfer (Krishna et al., 2020) (resulting paraphrases adhere to a certain language style). To increase paraphrase diversity, syntax control approaches can be used (Goyal and Durrett, 2020; Chen et al., 2020). Promising results have been achieved in zero-shot settings to produce paraphrases in a multi-lingual domains (Thompson and Post, 2020), or finetuning using only prompt tuning and Low Rank Adaptation (Chowdhury et al., 2022). These previous works showed the capabilities of LLMs to produce diverse and valid paraphrases, albeit with various syntax or semantic restrictions and additional mechanisms needed to produce good results.

In a number of recent studies, ChatGPT [1] has been applied to a variety of tasks. It has been used as a mathematical solver where it achieves below average human performance (Frieder et al., 2023), sentiment analysis task with varying performance (Zhu et al., 2023) and also as a general NLP task solver where it shows good performance on reasoning tasks (Qin et al., 2023). Another study (Wang et al., 2023) measured ChatGPTs capabilites to evaluate outputs of several natural language generation tasks, achieving performance of human evaluators. ChatGPT performs well in translation of high resource languages (Jiao et al., 2023). As for some other typical crowdsourcing tasks, ChatGPT outperforms the crowd in classification of political affiliation of Twitter users (Törnberg, 2023) and in-text annotation tasks (Gilardi et al., 2023). When (zero shot) ChatGPT is compared with a finetuned BERT, ChatGPT falls short in detection of paraphrases and text similarity tasks, but performs well on sentiment analysis, QA and inference (Zhong et al., 2023). These results indicate good capabilities of ChatGPT for NLP tasks, and its potential to replace crowd workers in at least some tasks (Törnberg, 2023; Gilardi et al., 2023).

## 3 ChatGPT paraphrase validity and diversity

To test whether ChatGPT can be an alternative to crowd workers in paraphrase generation, we replicated the data collection process of (Larson et al., 2020), who crowdsourced paraphrases to create datasets for the intent classification task. In their study, workers created paraphrases of seed sentences (with known intents). We replicate their process using ChatGPT and show that ChatGPT generated data are more diverse both in terms of lexical and syntactical diversity than those created by humans.

### 3.1 Data collection using ChatGPT

Larson et al. (2020) crowdsourced paraphrases for 10 intent classes (centered on the personal finances domain), using 3 sentences per class as seeds. For each given seed, a worker was asked to provide 5 paraphrases. Interested in increasing the resulting data diversity (and the downstream intent classification model robustness), Larson et al. collected data in two modes (to compare their outcomes):

---

[1] https://openai.com/blog/chatgpt

Figure 1: The paraphrases were created in three rounds, using two modes of worker prompting. Five datasets created were combined into two final datasets (prompt and taboo) used for further comparisons.

1. *Prompt*, where only seed sentence was shown to a worker (along with the instructions to paraphrase it).

2. *Taboo*, where prompt was shown along with a list of taboo words that workers should avoid when paraphrasing. The taboo words were selected from words overrepresented in previous paraphrases.

Larson et al. collected data in three rounds, illustrated in Figure 1. In the first round, workers created paraphrases using *Prompt* only method (since no taboo words could be known at that stage). The second and third rounds were conducted in *Prompt* mode (same instructions as the first round) and also in *Taboo* mode, where taboo words were determined based on previous rounds. The five resulting datasets were then combined into two: *prompt dataset* (which included all paraphrases gathered via Prompt mode) and *taboo dataset* (which included Prompt mode data from the first round and Taboo mode data from the second and third rounds).

In our own study, we replaced the crowd workers with ChatGPT and retained as much as possible from the original protocol of Larson et al. Specifically, we kept the three rounds and two modes of data collection and also created two resulting datasets (Prompt GPT and Taboo GPT). We used the same intent classes and seed sentences. We collected similar number of samples (see Table 1). As a slight protocol alteration, we chose to use the same taboo words as Larson et al. (instead of computing our own based on the first, resp. second collection rounds). We did this to allow better comparison between Taboo human and Taboo GPT datasets [2].

Data collection was carried out on 5 May 2023, using the *gpt-3.5-turbo-0301* model checkpoint. Our instructions to ChatGPT followed the general outlook of prompts given to workers in the original study. The ChatGPT prompt itself had this structure: "*Rephrase an original question or statement 5 times. Original phrase: [seed phrase]*". Along with the prompt, ChatGPT also accepts a "system" message which we specified as follows: "*You are a crowdsourcing worker that earns a living through creating paraphrases.*" This was to enable ChatGPT to better play the paraphraser role. For the Taboo mode the same prompt was used, amended with the following sentence: "*Don't use the words [taboo_1], [taboo_2] , ..., or [taboo_n] in your responses.*", where *taboo_n* represents the taboo words for the given seed. As in the original study, we used 3 taboo words for the second round and 6 taboo words for the third round.

To execute requests to ChatGPT, we used the chat completion API [3]. The entire example request can be seen in Appendix A. As for the other parameters, we set temperature to *1* (the higher the value the more randomness in the output), n to *13* (number of returned responses), model to *gpt-3.5-turbo* and the presence penalty to *1.5* (positive values penalize new tokens based on their existing frequency in the text so far, decreasing the model's likelihood to repeat the same line). Other parameters were kept to their default values. The parameters remained the same for all iterations and variations of the data collection.

In one round of data collection, we collected paraphrases for each of the 10 intents, with 3 seed phrases each. For each seed, we gathered 13 unique responses, expecting 5 different paraphrases in

---

[2]*https://github.com/kinit-sk/Crowd-vs-GPT-intent-class*

[3]https://api.openai.com/v1/chat/completions

each. For one data collection round, we collected 1950 paraphrases from ChatGPT. This results into 5850 paraphrases for both the `Prompt` and `Taboo` dataset (including duplicates and non-valid paraphrased, which are detailed in the next section).

## 3.2 ChatGPT data characteristics and validity

We analyzed the ChatGPT-generated paraphrases and compared them against the crowdsourced data from the original study. To answer RQ1, we assessed the paraphrase validity.

First, we counted and filtered the duplicated solutions. *The ChatGPT generated data contain more duplicates than crowdsourced data* (by exact string matching). ChatGPT also generated a small number of blank strings (blank lines). From the original 5850 samples of the ChatGPT `Prompt` dataset we have found that 20 blank texts and 610 are duplicates. After cleaning, this resulted in 5170 unique paraphrases from ChatGPT collected using the `Prompt` mode. In comparison, the crowdsourced `Prompt` dataset from the original study had 6091 collected samples out of which 442 were duplicates, totaling 5649 samples. The ChatGPT `Taboo` dataset contained 10 blank paraphrases and 196 duplicates, totaling to 5608 valid samples. The crowdsourced `Taboo` dataset had 5999 samples with 58 duplicates, resulting in 5941 samples in total.

Next, we have manually evaluated the validity of all ChatGPT paraphrases, i.e. we checked whether they are semantically equivalent to the seed sentences and their intents. To validate the created paraphrases we used a simple web application that we developed for this task. The user, who was one of the authors, was shown the seed samples, from which ChatGPT generated the paraphrases, and one particular paraphrase to validate. The author was then able to either mark the paraphrase as valid or not, with an additional optional checkbox to label the paraphrase as 'borderline case' for possible revisions. As the seed sentence changed only once in a while (one time for each 5-20 paraphrases) this significantly reduced the cognitive load on the annotator. We also discussed the 'borderline cases' where the user was not sure about the validity of created paraphrases. *All paraphrases were valid* and intent-adhering, we therefore conclude the RQ1 with a positive answer with some limitations. We noticed some stylistic divergences in the created paraphrases. Approximately 5% of the times, the paraphrases created by ChatGPT are odd in their style (usually using a very formal language; see examples in Table 2). We also observed that ChatGPT refrains from using slang or non-valid grammatical forms. We speculate that this may be due to the "role" given to the model through its system message (see section 3.1) or due to the extended vocabulary often found in formal documents.

For the data acquired through `Taboo` mode we also checked if ChatGPT adheres to the taboo instruction. In the original study, the solutions that did contain the tabooed words were not included in the resulting data (we thus can't evaluate the crowd performance in this regard). *In ChatGPT Taboo data, we can see ChatGPT breaking the taboo word rule in a minority of cases.* For the first round of taboo data collection (2nd round overall), where 3 taboo words were given to ChatGPT per task, we found out that ChatGPT ignored the taboo instruction for 167 out of 1950 samples. In the second round of taboo data collection (3rd overall), where 6 taboo words were given to ChatGPT per task, ChatGPT ignored these instructions for 331 samples out of 1950. Following the original study protocol, we removed samples containing taboo words. This resulted in the final 5143 samples for the `Taboo` dataset collected via ChatGPT.

*ChatGPT-generated samples are on average longer than those collected from humans.* Visualization of the distributions of the number of unique words and lengths of generated samples can be found in Appendix B in the Figure 3.

## 3.3 Diversity of ChatGPT paraphrases

To answer RQ2, we measured the lexical and syntactical diversity of the collected datasets. Our findings are summarized in Table 1.

We evaluated the *lexical diversity* in the same way as in the original study: by vocabulary size of different datasets. From this point of view, the smallest vocabulary (number of unique words) can be observed in the crowdsourced `prompt` mode data with 946 unique words. Compared to this, the ChatGPT `prompt` mode data features 1218 unique words, which is a 28.75% increase in vocabulary size. The crowdsourced dataset collected through the `taboo` mode had even higher vocabulary size with 1487 unique words. However, the ChatGPT `taboo` mode beats it even more with 1656 unique words (an increase of 11.37%). We conclude that *data collected via ChatGPT has higher lexical diversity* compared to crowdsourced data when the

| Dataset type | # collected samples | # after filtering | # unique words ($\uparrow$) | $\overline{TED}$ ($\uparrow$) |
|---|---|---|---|---|
| Prompt human | 6091 | 5649 | 946 | 13.686 |
| Taboo human | 5999 | 5941 | 1487 | 15.483 |
| Prompt Falcon | 5850 | 2897 | 810 | 14.382 |
| Taboo Falcon | 5850 | 1646 | 643 | 25.852 |
| Prompt GPT | 5850 | 5170 | 1218 | 19.001 |
| Taboo GPT | 5850 | 5143 | 1656 | 18.442 |
| Taboo GPT+tab. samples | 5850 | 5608 | 1871 | 18.661 |

Table 1: The datasets used for comparisons in this work. The data originally crowdsourced by Larson et al. are denoted "human", while data collected in our work are denoted GPT and Falcon. The GPT data have higher lexical and syntactical diversity than human data (within collection modes) and contain slightly more duplicates, while the Falcon data contained a lot of invalid samples. Using taboo mode increases the no. unique words for GPT data with the up arrow indicating 'the higher the better'.

same data collection mode is used. This can also be seen on the visualization of the number of unique words in Figure 3.

We also compare the collected datasets on *syntactical diversity* to better assess the structural variations in paraphrases. We do this by calculating the *tree edit distance* value (TED) (Chen et al., 2019) between all pairs of paraphrases sharing the same intent. TED is first calculated pair-wise for each phrase in the intent and data split (separate for human, GPT, original). This is then averaged to get the mean – this should represent how syntactically diverse the phrases are - the higher the number of mean TED is, the more diversity is present. When comparing prompt datasets, ChatGPT created more diverse sentence structures with a mean TED value of 19.001 compared to a 13.686 mean TED value for crowdsourced data. The same holds for the taboo datasets: the crowdsourced taboo dataset has a mean TED value of 15.483 while the ChatGPT collected dataset has 18.442. It should be noted that while the data collected using human workers have higher TED value for the taboo method than for the prompt method, the same cannot be said about the data generated from ChatGPT - the introduction of taboo words to ChatGPT does not increase the syntactical diversity. We have confirmed our findings by running a Mann-Whitney-U test between datasets with $p = 0.001$ for the measured TED values. We conclude that *data collected via ChatGPT has higher syntactical diversity than that collected from human workers* for the same data collection method.

Turning back to RQ2, we therefore conclude that ChatGPT generates more diverse paraphrases than crowd workers.

## 3.4 Comparison of ChatGPT paraphrases with Falcon

As ChatGPT is a closed model, we sought to compare its capability to produce paraphrases with an open source LLM. For this, we selected Falcon-40B-instruct [4] as one of the leading open source LLMs at the time of writing of this paper. We did not conduct any specific parameter tuning, used the same parameter values as for ChatGPT, collected the same amount of data as with ChatGPT and used the same prompts as we did for the ChatGPT experiments (with only minimal alterations required by the model). Our findings are summarized in Table 1.

We preprocessed the data - removed duplicates and for Taboo split removed paraphrases which contained tabooed words. This resulted in 3784 samples for the Prompt split and 2253 samples for the Taboo split. When compared with results of ChatGPT collected data in Table 1, the Falcon collected data contain more duplicates and invalid sentences, which resulted in fewer samples.

Next, we manually annotated the collected data with the same process as explained above. Compared to ChatGPT, where we detected no paraphrases that had altered meaning, the Falcon collected data was of considerably lower quality. The Falcon model struggled to produce paraphrases and follow the basic instructions at times: the created paraphrases were often meta-comments (e.g. '*As an AI language model..*'), missed the intent of the seed paraphrase or were responses to the seed paraphrases (e.g. from seed '*When is the bank open*' the response was '*Which bank do you mean?*').

---

[4]https://huggingface.co/tiiuae/falcon-40b-instruct

| Intent | ChatGPT generated phrases | Odd ChatGPT generated phrases |
|--------|---------------------------|-------------------------------|
| phone | *How can I connect with my bank over the phone?* | *Is it possible to share the telephone directory of my financial establishment with me?* |
| location | *Can you guide me to the nearby location of a bank?* | *Which area has an institution mainly dedicated to handling money matters?* |
| balance | *Can you tell me my financial status?* | *What is the procedure to review my remaining funds?* |
| safe | *Could you help me find a way to keep my valuable items at the bank?* | *Are there any options available for me to keep my precious belongings in a protected setting at the banking establishment?* |
| hours | *What is the hour when the bank starts operating?* | *At what moment do the bank employees arrive to commence their workday?* |

Table 2: Examples of collected phrases for the *Taboo* method via ChatGPT. We list some typical phrases created by ChatGPT, as well as some "odd" phrases.

This resulted in future filtering of the datasets per split. For the `Prompt` split we removed 887 (23.44%) samples resulting in 2897 valid samples and for the `Taboo` split we removed 607 (26.94%) samples resulting in 1646 valid samples. Finally, we computed the no. unique words in the splits with 810 unique words in the `Prompt` split and 634 unique words in the `Taboo` split and the TED value for the `Prompt` split was 14.382 and 25.852 for the `Taboo` split.

The higher TED values (or higher syntactical diversity) and lower lexical diversity for `Taboo` split when compared to the `Prompt` split on Falcon-collected data can be interpreted to be due to the lower amount of valid samples after filtering in the `Taboo` split, where only a limited vocabulary with a lot of syntactical variety is present.

In conclusion, when compared to ChatGPT-collected data, the Falcon-collected data yielded more duplicates and invalid samples that resulted in a lower quality dataset in terms of lexical diversity. For this reason, we only use ChatGPT in model robustness experiments described further. However, the performance of open source models is increasing and future models may outperform ChatGPT, giving an interesting outlook for future comparisons.

## 4 Model robustness

To compare the robustness of models trained on ChatGPT and human paraphrases (RQ3), we computed accuracy of these models over out-of-distribution (OOD) data. The comparison indicates higher or comparable accuracy of ChatGPT-data-trained models.

### 4.1 Data and models used in the experiment

The data we used for diversity evaluation (section 3) did not include suitable OOD test data for robustness tests. Therefore, we turned to a set of 5 existing benchmark datasets for intent classification. As part of their work, Larson et al. sampled these 5 datasets for seeds to create paraphrases through crowdsourcing. As previously, we replicated the data collection using ChatGPT instead. The seeds used in this data collection constitute only a fraction of the original 5 datasets: therefore, we could take the remainder to act as our OOD test data.

The datasets were following: ATIS (Hemphill et al., 1990), Liu (Liu et al., 2021), Facebook (Gupta et al., 2018), Snips (Coucke et al., 2018) and CLINC150 (Larson et al., 2019b). The intent domains and coding schemes varied from dataset to dataset and included restaurants, flight booking, media control, and general knowledge. Only intents included by seed samples were used for evaluation – some intents were thus omitted. The seed samples were selected by Larson et al. The total no. samples for each dataset can be found in Appendix C.

The paraphrase collection procedure was similar to one of diversity evaluation (section 3), with some key differences. There were 5 sets of seeds (one per each dataset). For each seed set, data was collected in four rounds (instead of three). In each round ChatGPT was provided with the same seed set to paraphrase. The iterations differed by the number of taboo words (0, 2, 4 and 6). The resulting paraphrases were merged into one final paraphrase dataset (per each of the five benchmark dataset used). Empty responses, duplicates and

paraphrases with taboo words were filtered out[5]. As previously, we manually checked the validity of all paraphrases.

From now on, we denote the crowdsourced paraphrases from the study of Larson et al (Larson et al., 2020) as the `human` data. We denote the ChatGPT collected data as the `GPT`. The OOD test data (the "non-seed" remainder of the original 5 datasets) are denoted as the `original` data. Figure 2 illustrates the evaluation process for one benchmark dataset.

We have also checked for possible overlaps between `GPT`, `human` and `original` data for all the datasets to check if crowdsourcing workers or ChatGPT have not generated the same sentences as are included in the `original` data. We have detected less then 1% of collected data to be overlapping and have removed it from the experiments. The results and details can be found in the Appendix E.

### 4.2   Accuracy on out-of-distribution data

We evaluate our experiments for two models: we fine-tuned BERT-large [6] for 5 epochs using the huggingface library and we fine-tuned the SVM [7] multiclass classifier with TF-IDF features using the standard sklearn implementations.

We report our results for BERT in Table 3 while the results for the SVM classifier can be found in the Appendix F. As the results show, models trained on ChatGPT (`GPT`) generated datasets achieve comparable results to models trained on `human` data when evaluated on the OOD `original` data in 2 cases (Liu and Snips) and outperform the models trained on `human` data in 3 cases (Facebook, CLINC150 and Snips datasets). Similar effects can be observed for the SVM classifier as well. This indicates that models trained on `GPT` data have equal (or in some cases better) robustness than models trained on `human` data for the intent classification task. Full results for models trained on each dataset can be found in Appendix F.

### 5   Cost comparison: ChatGPT vs crowdsourcing?

We compare the crowdsourcing costs of the original study with our ChatGPT replication and we do this for both the diversity experiments and for the model robustness experiments on OOD data. In all of the experiments the authors of the original study estimate the cost of one paraphrase to be at $0.05. The pricing of the API during the collection of these samples was $0.002 per 1K tokens.

For diversity experiments, the number of collected paraphrases from the original study is 10,050 samples. This results in an approximate cost of the original study at approximately $500. We have collected a total of 9,750 samples from ChatGPT. This resulted in a total cost of our data collection experiments of approximately $0.5. This is a 1:1000 ratio for ChatGPT when both studies are compared.

For the model robustness experiments, the total data size from the original study for all the 5 benchmark datasets combined is 13,680 samples, which corresponds to an approximate cost of the original study at $680. In our experiments, we have collected 26,273 samples for $2.5. This results in an approximate ratio of 1:525 in favor of ChatGPT.

Together, both experiments make up the price ratio of 1:600 for ChatGPT. We conclude that the price of using ChatGPT for collecting paraphrases for intent classification is much lower than with crowdsourcing.

### 6   Discussion

Given the results of our experiments, we perceive ChatGPT as a viable alternative to human paraphrasing. However, we also will not argue for a complete replacement yet (for that, many more investigations are needed). ChatGPT, while generally powerful, still has some weaknesses.

During the manual validation of ChatGPT generated data we noticed that ChatGPT does not change named entities for their acronyms (or vice versa) or for their other known names. For example, the word 'NY', even though it refers to 'New York' in the context of the sentence, will always remain the same in all the paraphrases generated by ChatGPT. This would indicate that while ChatGPT is capable of producing very rich sentences in terms of lexical and syntactical diversity (as seen in Section 3.3), it does not produce alternative names for named entities, e.g. locations, songs, people, which is something crowdsourced data handles well.

The tendency of ChatGPT to produce "odd" paraphrases also has potential implications: the divergence from typical human behavior can harm many applications. On the bright side, oddity can indicate easier machine-generated text detection, which may serve (crowdsourcing) fraud detection.

---

[5]We have also investigated the effects of including the taboo paraphrases, see Appendix D

[6]https://huggingface.co/bert-large-uncased

[7]https://scikit-learn.org/stable/modules/generated/sklearn.svm.SVC.html

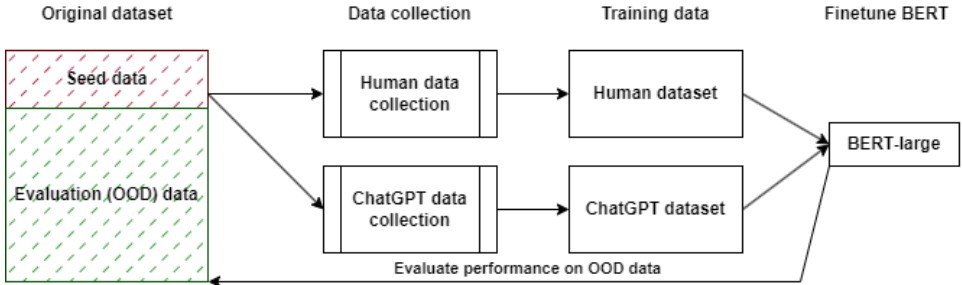

Figure 2: The evaluation process on out-of-distribution (OOD) data for one dataset on BERT-large, same process was repeated for SVM. This process has been repeated for all the datasets for a comparison of robustness for models trained on ChatGPT and human collected data.

| | | FB | ATIS | Liu | CLINC150 | Snips |
|---|---|---|---|---|---|---|
| Human | # samples | 3092 | 2302 | 1140 | 3019 | 4216 |
| | accuracy | 72.65(1.24) | 79.46(1.83) | **93.81**(0.98) | 95.42(1.04) | 98.89(0.23) |
| | conf. interval | [71.41-73.89] | [78.33-80.59] | [93.02-94.59] | [94.59-96.26] | [98.72-99.08] |
| GPT | # samples | 2976* | 2210* | 1133* | 3194* | 3961* |
| | accuracy | **76.53**(2.71) | **87.64**(3.26) | 93.55(0.41) | **98.06**(0.30) | **99.13**(0.18) |
| | conf. interval | [74.85-78.21] | [85.62-89.67] | [93.21-93.88] | [97.82-98.29] | [98.98-99.27] |

Table 3: Accuracy of a BERT language models over 10 runs, finetuned for the task of intent classification for different datasets on the GPT data and human data from the original study. Values in brackets near the mean denote the standard deviation. We also report the number of samples per each dataset for each dataset, with the asterisk meaning that the dataset was downsampled and the 95% confidence intervals for accuracy. Models trained on GPT data have better robustness on OOD data in 3 cases and comparable robustness in 2 cases when compared to models trained on human data.

As our results also show, models trained on GPT data do not always perform better than those trained on human data. It should be noted that the OOD test data is imbalanced between labels in the case of FB, ATIS and Liu, although the number of samples for each label in the training data (both GPT and human) is balanced for each of the 5 datasets. This performance might be due to a number of factors. First, the number of samples used for training in the Liu dataset is the smallest of all datasets, which might result in a distribution bias where not enough data was collected to have a better comparison between models trained on GPT and human data. Second, the lack of paraphrases or alternative names for named entities in the GPT data might result in poorer performance if the test data contain many alternative names. Third, there may be other hidden biases in the datasets.

All of these observations indicate that ChatGPT is able to create diverse paraphrases both in terms of their structure and vocabulary, but with certain limitations. Given the current state of our knowledge, we see ChatGPT as a paraphrase creator especially for data augmentation scenarios, teamed

up with human paraphrasers and overseers.

## 7 Conclusion

In this work, we compared the quality of crowd-sourced and LLM generated paraphrases in terms of their diversity and robustness of intent classification models trained over them. We reused the data collection protocol of the previous crowdsourcing study of Larson et al. (2020) to instruct ChatGPT and Falcon-40B instead of crowd workers. Our results show that ChatGPT collected data yield higher diversity and similar model robustness to the data collected from human workers for a fraction of the original price, while Falcon-40B struggled in creating valid and unique paraphrases. The effects of human-collected and ChatGPT-collected data on robustness vary, which might indicate the need for their combination for best model performance.

The much cheaper ChatGPT data collection provided us with a better dataset (in many aspects). Given our observations, it does appear beneficial to us to use ChatGPT as a supplement to crowdsourcing for paraphrase collection. We do not rule out the use of human computation for this task, but

at the same time, we speculate that the usage of ChatGPT as a data enhancement method might be beneficial for dataset building.

Our results lead to new questions about ChatGPT's capabilities for the creation of paraphrases. A possible point of diminishing returns for this paraphrasing task in ChatGPT might be explored in order to determine when too many duplicates are created from ChatGPT. The usage of different diversity incentives on ChatGPT could further improve the performance of ChatGPT. Another area of interest might be the usage of ChatGPT paraphrasing for low-resource languages, where further investigation to the quality of created data is needed.

## 8   Limitations

There are multiple limitations to our current work.

First, we did not estimate when the point of diminishing returns for ChatGPT paraphrasing happens - how many times can we collect paraphrases for the same seed sentence before ChatGPT starts producing too many duplicates. Our experiments show, that taboo words and other similar methods could mitigate this, as the number of duplicates is much smaller when such restrictions are used. However, it might be the case that a large enough crowd could beat ChatGPT in paraphrasing for the same seed in terms of diversity, most probably when ChatGPT starts repeating itself.

Second, ChatGPT does not paraphrase or use alternative names for named entities such as locations, people, songs, etc. This might be mitigated with prompt engineering, but this currently limits its paraphrase outputs in cases where named entities are present in seed sentences.

Third, we have not used any specific prompt engineering during our work, which might produce better results for ChatGPT, nor have we investigated in depth the effects of different API parameters on the created paraphrases.

Fourth, we performed experiments for only one language, namely English, while ChatGPT can be used to produce paraphrases in other languages. As such, we do not know what quality of paraphrases would this model produce for different langauges.

Fifth, we have not compared the performance of ChatGPT with other LLMs such as Alpaca [8], Vicuna [9] or LLaMa (Touvron et al., 2023).

---

[8] https://crfm.stanford.edu/2023/03/13/alpaca.html
[9] https://lmsys.org/blog/2023-03-30-vicuna/

Sixth, we have not further investigated the source of the mixed results in Section 4 and have only speculated at the source of these uneven results.

Seventh, the reproducibility of our data collection process is dependent upon the owners of ChatGPT services - the models get further finetuned with new data and other changes are made to them without the models themselves being public. This means that there is no guarantee that our study can be reproduced in an exact manner in the future.

Eigth, the performance of Falcon-40B-instruct in Section 3.4 could be enhance via further fine-tuning of the model, possibly yielding better results.

Ninth, the good results for model robustness on OOD data of ChatGPT-collected data can be attributed to the data on which ChatGPT was trained: it is possible that ChatGPT has been trained with a corpus containing all the datasets used in our evaluation. Nevertheless, ChatGPT is able to create lexically and syntactically rich paraphrases that lead to good generalization for models trained on such data while generally avoiding recreating the same sentences that were in those datasets as can be seen in Section 4.1.

## Ethics Statement

We see no ethical concerns pertaining directly to the conduct of this research. In our study, we analyzed existing data or data generated using ChatGPT API. Albeit production of new data through LLMs bears several risks, such as introduction of biases, the small size of the produced dataset, sufficient for experimentation is, at the same time, insufficient for any major machine learning endeavors, where such biases could be transferred.

Our findings, of course, highlight concerns for the future: the potential of LLMs to replace humans in some *human intelligence tasks*. A threat to crowdsourcing practices as they have been known prior 2022 is already materializing: a recent study of Veselovsky et al. (2023) measured the probable use of LLMs by crowdworkers in 33-46% cases.

## Acknowledgements

This research was partially supported by the Central European Digital Media Observatory (CEDMO), a project funded by the European Union under the Contract No. 2020-EU-IA-0267, and has received funding by the European Union under the Horizon Europe vera.ai project, Grant Agreement number 101070093.

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

## A   Example code for sending requests to the ChatGPT API

We have collected paraphrases from ChatGPT using similar code as below (the same API parameters were used in our data collection):

```python
def request_response_from_gpt(prompt):
    response = openai.ChatCompletion.create(
    model="gpt-3.5-turbo",
    messages=[
    {"role": "system", "content": "You
    are a crowdsourcing worker that
    earns a living through creating
    paraphrases."},
    {"role": "user", "content": prompt
    }],
    temperature=1,
    frequency_penalty=0.0,
    presence_penalty=1.5,
    n=13)
    return response
```

## B   Further visualization of the collected data

A more in-depth insight into characteristics of the collected datasets (number of words and content of

paraphrases) is provided by Figure 3 and Figure 4.

# C  Datasets used in OOD experiments

This section provides an overview of datasets used in Section 4. We used the same samples and intents as in the original study (Larson et al., 2020) for all the datasets for a direct comparison with human collected data. Samples that were not used as seed samples for the data collection were used during the evaluation as seen in Table 3.

Facebook dataset is an intent classification and slot filling dataset with intents about interaction with a virtual assistant (Gupta et al., 2018). As per the original study (Larson et al., 2020), we used 10 intents and for each intent 30 queries were seed samples for the data collection itself. The used intents were: get_directions, get_distance, get_estimated_arrival, get_estimated_departure, get_estimated_duration, get_info_road_condition, get_info_route, get_info_traffic, get_location and update_directions.

Snips dataset is intent classification and slot filling dataset (Coucke et al., 2018) for which, as per the original study, 7 intents were used. For each intent we used 50 samples as seed samples for the data collection. The used intents were: PlayMusic, AddToPlaylist, BookRestaurant, GetWeather, RateBook, SearchCreativeWork and SearchScreeningEvent.

ATIS corpus is a benchmark dataset (Hemphill et al., 1990) used for slot-filling and intent classification with intents related to interaction with a flight booking assistant. Per the original study, 6 intents were sampled and for each of them 8 to 50 samples were sampled to be used as seeds for the data collection. The used intents were: atis_abbreviation, atis_aircraft, atis_airfare, atis_airline, atis_flight and atis_ground_service.

Liu dataset was used as an intent classification benchmark. Intents are similar to the Facebook and Snips datasets and, as per the original study, 10 intents were sampled with 10 samples each that were used for the data collection. The used intents were: cooking_recipe, datetime_query, audio_volume_up, news_query, audio_volume_down, weather_query, qa_currency, play_music, transport_traffic and music_query.

CLINC150 dataset is an intent classification benchmark with a variety of different intents. 40 intents were sampled and for each intent 10 queries were used as seed data for the data collection process.

## C.1  Dataset statistics for OOD experiments

We report the number of samples in each dataset after filtering out unrelevant intents and data cleaning as per Section 4. In our experiments we always downsampled datasets each run to adjust for the different number of samples in each dataset.

For the Liu dataset experiments the human data contains 1140 samples, the GPT data contains 2354 with taboo samples and 2265 samples without those samples and the original data contains 4171 samples in the train and 1087 samples in the test split.

For the Facebook dataset experiments the statistics are: human data contains 3092 samples, the GPT data contains 6171 with taboo samples and 5937 samples without those samples and the original data contains 19398 samples in the train and 5645 samples in the test split.

For the ATIS dataset experiments the statistics are: human data contains 2303 samples, the GPT data contains 4654 with taboo samples and 4420 samples without those samples and the original data contains 3985 samples in the train and 759 samples in the test split.

For the CLINC150 dataset experiments the statistics are: human data contains 3019 samples, the GPT data contains 4767 with taboo samples and 4365 samples without those samples and the original data contains 3500 samples in the train and 969 samples in the test split.

For the Snips dataset experiments the statistics are: human data contains 4126 samples, the GPT data contains 8327 with taboo samples and 7922 samples without those samples and the original data contains 13615 samples in the train and 697 samples in the test split.

# D  Model robustness on OOD data with the inclusion of taboo samples and model training details

We report the results of OOD evaluation for SVM model with TF-IDF features in Table 4 for both data with and without taboo samples and the results for BERT-large with taboo samples in Table 5. The inclusion of taboo samples during training leads to more robust models trained on GPT data.

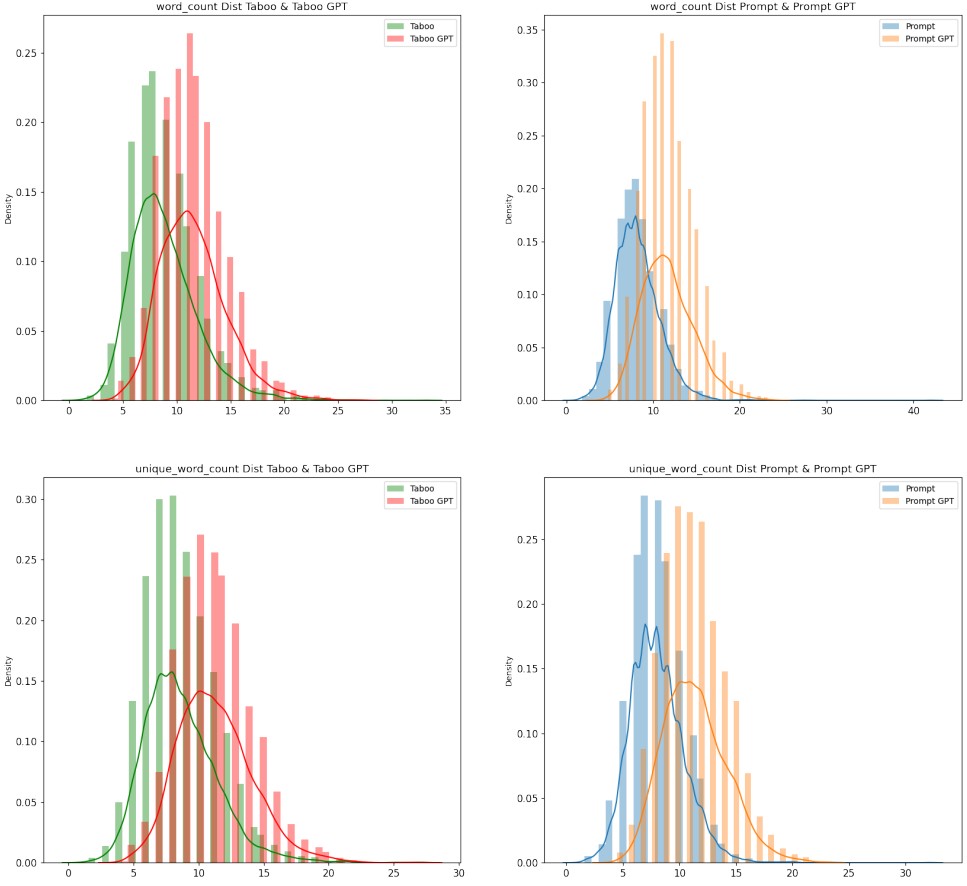

Figure 3: A comparison of the GPT and human collected data in terms of no. words in paraphrases in row 1 and no. unique words in paraphrases in row 2. The GPT-generated paraphrases are longer and have more unique words.

For BERT finetuning we used 2 or 4 training batch size, 16 or 32 evaluation batch size and *1e-5* learning rate finetuned for 5 epochs. For training we used a machine with 1x NVIDIA GeForce RTX 3090 24GB GPU, 32 GB RAM and 8 CPU cores. Two training sessions were ran in parallel on the same GPU (one with 2 and second 4 training batch size), for a total of 10 training runs for each of the 5 datasets.

## E   Overlaps between data on the 5 different datasets

To determine overlaps between different data (`original`, GPT and human) on 5 different datasets, we lemmatized all of the texts, casted to lowercase and removed any punctuation. The results can be found in Table 6. GPT data have less overlaps on `original` than human data have.

## F   Full results of model robustness for `original`, GPT and human data on 5 different dataset

We report the full results of models trained on `original` datasets, GPT data and human data as per Section 4 for each dataset in Table 7 for BERT and in Table 8 for SVM. As can be seen in both Tables, models trained on `original` data have a considerable drop in accuracy for GPT and human data, while models trained on GPT data achieve the best results in terms of robustness.

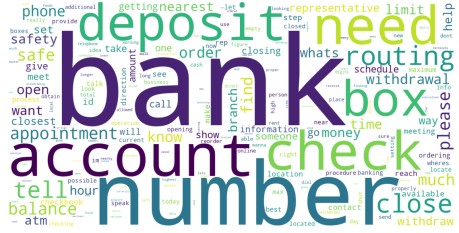

(a) Word cloud visualization of the `Prompt human` dataset.

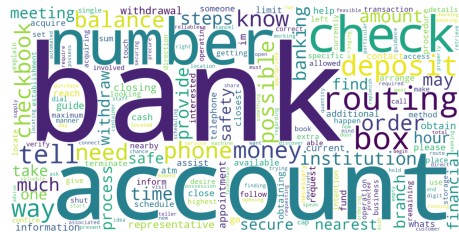

(b) Word cloud visualization of the `Prompt GPT` dataset.

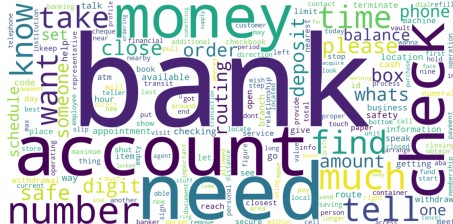

(c) Word cloud visualization of the `Taboo human` dataset.

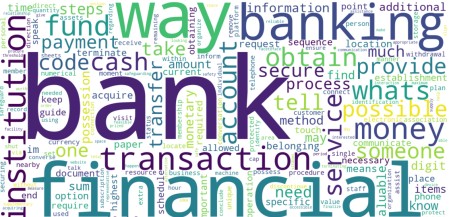

(d) Word cloud visualization of the `Taboo GPT` dataset.

Figure 4: Visualization of different word clouds from the collected data. The word clouds from ChatGPT data are more dense, with the general same most frequent words, although some differences are present (e.g. the word *financial* in Figure 4d.

| Train data split | Test original data (OOD) | | | | |
|---|---|---|---|---|---|
| | FB | ATIS | Liu | CLINC150 | Snips |
| Human | 72.65 | 79.46 | **93.81** | 95.42 | 98.89 |
| GPT | 76.53 | 87.64 | 93.55 | 98.06 | **99.13** |
| GPT + tab. samples | **79.64** | **87.74** | 93.13 | **98.42** | 99.07 |

Table 4: Accuracy of finetuned BERT-large over 10 runs, finetuned for the task of intent classification for different datasets on the ChatGPT collected data with and without taboo samples and human collected data from the original study. The inclusion of taboo samples in training generally leads to slightly increased robustness.

| Train data split | Test original data (OOD) | | | | |
|---|---|---|---|---|---|
| | FB | ATIS | Liu | CLINC150 | Snips |
| Human | **66.72** | 81.79 | 80.93 | 87.07 | **98.62** |
| GPT | 60.39 | 81.12 | 81.54 | 94.76 | 97.98 |
| GPT + tab. samples | 64.59 | **81.81** | **89.93** | **95.40** | 97.85 |

Table 5: Accuracy of an SVM with TF-IDF features over 10 runs, finetuned for the task of intent classification for different datasets on the ChatGPT collected data with and without taboo samples and human collected data from the original study. The inclusion of taboo samples in training generally leads to slightly increased robustness.

|        | FB |          |
|        | Human | Original |
|--------|-------|----------|
| GPT    | 10    | 0        |
| Human  | -     | 15       |

(a) Overlaps between different different data for the `Facebook` dataset. The cells denote the no. samples that are in both the data in the row and column.

|        | ATIS |          |
|        | Human | Original |
|--------|-------|----------|
| GPT    | 25    | 1        |
| Human  | -     | 24       |

(b) Overlaps between different different data for the `ATIS` dataset. The cells denote the no. samples that are in both the data in the row and column.

|        | Liu |          |
|        | Human | Original |
|--------|-------|----------|
| GPT    | 10    | 3        |
| Human  | -     | 8        |

(c) Overlaps between different different data for the `Liu` dataset. The cells denote the no. samples that are in both the data in the row and column.

|        | CLINC150 |          |
|        | Human | Original |
|--------|-------|----------|
| GPT    | 16    | 13       |
| Human  | -     | 24       |

(d) Overlaps between different different data for the `CLINC150` dataset. The cells denote the no. samples that are in both the data in the row and column.

|        | Snips |          |
|        | Human | Original |
|--------|-------|----------|
| GPT    | 19    | 0        |
| Human  | -     | 12       |

(e) Overlaps between different different data for the `Snips` dataset. The cells denote the no. samples that are in both the data in the row and column.

Table 6: Overlaps in no. samples between different data for each of the 5 datasets used in OOD data experiments. The GPT data have less overlaps on `original` data than human data.

|              | Test data |       |          |
|--------------|-----------|-------|----------|
| Train data   | GPT       | Human | Original |
| GPT          | **96.59** | 91.33 | 76.53 |
| Human        | 93.22 | **94.37** | 72.65    |
| Original     | 75.71     | 73.56 | **96.60** |

(a) Results on the Facebook dataset.

|              | Test data |       |          |
|--------------|-----------|-------|----------|
| Train data   | GPT       | Human | Original |
| GPT          | **98.65** | 95.19 | 87.64 |
| Human        | 95.87 | **96.93** | 79.46    |
| Original     | 77.06     | 65.76 | **99.89** |

(b) Results on the ATIS dataset.

|              | Test data |       |          |
|--------------|-----------|-------|----------|
| Train data   | GPT       | Human | Original |
| GPT          | **99.16** | 98.46 | 93.55    |
| Human        | 95.69 | **97.78** | 93.81 |
| Original     | 90.43     | 93.57 | **97.13** |

(c) Results on the Liu dataset.

|              | Test data |       |          |
|--------------|-----------|-------|----------|
| Train data   | GPT       | Human | Original |
| GPT          | **99.23** | 93.98 | **98.06** |
| Human        | 83.94 | **96.77** | 95.42    |
| Original     | 74.83     | 82.95 | 96.46 |

(d) Results on the CLINC150 dataset.

|              | Test data |       |          |
|--------------|-----------|-------|----------|
| Train data   | GPT       | Human | Original |
| GPT          | **99.45** | 96.70 | **99.13** |
| Human        | 97.36 | **98.91** | 98.89 |
| Original     | 92.25     | 94.07 | 98.78    |

(e) Results on the Snips dataset.

Table 7: Results for each dataset for BERT finetuned on differently collected data. We report the mean accuracy over 10 runs with different train/test splits each time. The best results for each column (test data) are in bold and the second best results are underlined. Models trained on original datasets tend to have the worst results in terms of robustness, while models trained on GPT data have the best results in terms of robustness.

|  | Test data | | |
| Train data | GPT | Human | Original |
|---|---|---|---|
| GPT | **96.32** | 92.34 | 60.39 |
| Human | 87.33 | **95.23** | 66.72 |
| Original | 38.39 | 58.71 | **94.65** |

(a) Results on the `Facebook` dataset.

|  | Test data | | |
| Train data | GPT | Human | Original |
|---|---|---|---|
| GPT | **97.76** | 96.25 | 81.42 |
| Human | 94.09 | **97.18** | 81.79 |
| Original | 34.14 | 38.98 | **91.96** |

(b) Results on the `ATIS` dataset.

|  | Test data | | |
| Train data | GPT | Human | Original |
|---|---|---|---|
| GPT | **97.61** | 93.08 | 81.54 |
| Human | 85.78 | **94.87** | 80.93 |
| Original | 51.92 | 63.47 | **91.71** |

(c) Results on the `Liu` dataset.

|  | Test data | | |
| Train data | GPT | Human | Original |
|---|---|---|---|
| GPT | **96.24** | 88.44 | **94.76** |
| Human | 75.33 | **93.44** | 87.07 |
| Original | 45.31 | 58.43 | *92.88* |

(d) Results on the `CLINC150` dataset.

|  | Test data | | |
| Train data | GPT | Human | Original |
|---|---|---|---|
| GPT | **99.68** | 99.15 | 97.99 |
| Human | 97.49 | **99.56** | **98.62** |
| Original | 68.10 | 85.42 | 97.99 |

(e) Results on the `Snips` dataset.

Table 8: Results for each dataset for SVM trained with TF-IDF features on differently collected data. We report the mean over 10 runs with different train/test splits each time. The best results for each column (test data) are in bold and we underline also the second best results. Similar to BERT, SVM models trained on `original` datasets tend to have the worst results in terms of robustness, while models trained on GPT data have the best results in terms of robustness.