# OpenReview forum: "ChatGPT to Replace Crowdsourcing of Paraphrases for Intent Classification: Higher Diversity and Comparable Model Robustness"
_EMNLP/2023/Conference — EMNLP 2023 Main_

### Official Review · Reviewer_fC6f · 2023-07-29

**Soundness:** 3

**Excitement:**

3: Ambivalent: It has merits (e.g., it reports state-of-the-art results, the idea is nice), but there are key weaknesses (e.g., it describes incremental work), and it can significantly benefit from another round of revision. However, I won't object to accepting it if my co-reviewers champion it.

**Paper Topic And Main Contributions:**

This paper empirically studies whether ChatGPT can replace the crowdsourcing of paraphrases for intent classification. Through replicating the existing study (Larson et al., 2020) with ChatGPT rather than human workers, this paper finds the ability of ChatGPT to yield higher diversity and similar model robustness than humans in collecting data for intent classification.

**Questions For The Authors:**

Question A: How do you design your parameters for using ChatGPT API? Specifically for example, why do you choose the temperature to 1 which means the output is random?

Question B: What is the arrow(↑)'s meaning which is shown after `TED` in Table 1?

**Reasons To Accept:**

1. This paper shows the ability of ChatGPT to yield higher diversity and similar model robustness than humans in collecting data for intent classification.
2. This paper leads to new questions about ChatGPT's capabilities for the creation of paraphrases.

**Reasons To Reject:**

1. The investigation in this paper is simple and not deep. For example, the scope of this paper is limited to the task of intent classification while ChatGPT has a wide range of abilities in crowdsourcing of paraphrases. Only ChatGPT is used for the investigation while many other open-source LLMs exist. The influence of parameters of using ChatGPT for this task is not profoundly investigated.
2. The writing of this paper is not very clear. Some of the unclear details are explained in the question.

**Reproducibility:**

3: Could reproduce the results with some difficulty. The settings of parameters are underspecified or subjectively determined; the training/evaluation data are not widely available.

**Reviewer Confidence:**

3: Pretty sure, but there's a chance I missed something. Although I have a good feel for this area in general, I did not carefully check the paper's details, e.g., the math, experimental design, or novelty.

**Typos Grammar Style And Presentation Improvements:**

The writing of this paper is a bit colloquial. The tense of some sentences is inconsistent.
For example,
line 281, "We also observed that ChatGPT ...": Past Tense
line 333, "We also compare ...": Present Tense

---

> ### Author Rebuttal · Authors · 2023-08-28
>
> Thank you very much for your assessment. Below, we address each question (or critical remark) individually.
>
> > The investigation in this paper is simple and not deep.
>
> We tried to keep the investigation methodology simple, yet still sufficient, to answer our research questions (which it did). We believe that the impact of LLMs on crowdsourcing is a new topic and therefore has basic questions open. Furthermore, simple methodologies are easier to reproduce.
>
> It would be a different situation, if the outcomes of the simple analysis were inconclusive or with marginal effects. Then, more in depth analysis (more metrics, content segmentation or qualitative content analysis) would be relevant. But we believe this is not the case.
>
> > For example, the scope of this paper is limited to the task of intent classification while ChatGPT has a wide range of abilities in crowdsourcing of paraphrases.
>
> Our paper investigates the general ability of ChatGPT to paraphrase to a large degree. The RQ1 (paraphrase validity) and RQ2 (paraphrase diversity) do just that. These RQs (measures) would be relevant for most (if not all) other downstream tasks utilizing paraphrases, not just for intent classification.
>
> The writing (paper title, abstract, intro) puts a lot of emphasis on the intent classification (to prevent unnecessary over-generalizing interpretations), however, that does not counter the real scope of the paper.
>
> Furthermore, the selection of the downstream task was mostly induced by the existing study which we replicated. From that study, we selected all aspects of the crowdsourcing outcomes that could reasonably be compared with ChatGPT performance.
>
> > Only ChatGPT is used for the investigation while many other open-source LLMs exist.
>
> The focus of our current paper was to compare the crowd vs. some widely known and used LLM to investigate if such LLMs could pose a threat to crowdsourcing by producing high quality (valid and diverse) paraphrases. We have therefore selected the most popular and accessible LLM at the time - ChatGPT. As its performance outperformed humans in our experiments, it sufficiently answered our research questions (without the need of investigating further LLMs). However, investigation of paraphrasing capabilities of other LLMs is a logical step to take and a future work for us.
>
> Prompted by the review(s), **we conducted new experiments with both Falcon-7B-instruct and Falcon-40B-instruct** (highest ranking models on the Huggingface benchmark at the time of submission of the paper). We will incorporate the results into the final version of the paper.
>
> We used the same setup as in the diversity experiments - replicating the method of Larson et al. 2020 for collecting paraphrases for 10 different financial intents. We did not conduct any specific parameter tuning, used the same parameter values as for ChatGPT and used the same prompts as we did for the ChatGPT experiments.
>
> We preprocessed the data - removed duplicates and for *taboo* split removed paraphrases which contained tabooed words. Next, as we did not have time to fully validate ~5.5k paraphrases, we randomly chose 600 paraphrases for each split per model to be annotated for validity (is this a valid paraphrase given intent and seed sentence?). The 4 annotated datasets were Falcon-7B prompt and taboo; and Falcon-40B *prompt* and *taboo*. The randomly chosen 600 paraphrases per dataset were annotated by one of the authors.
>
> Summary of our findings:
>
> - Both Falcon models struggled with producing a unified format for the paraphrases, resulting in a some additional overhead in parsing the results;
> - Both Falcon models struggled to produce paraphrases and follow the basic instructions at times.
> - For the Falcon-7B collected *prompt* data, 92 out of the 600 (15.33%) sampled paraphrases were invalid and for the *taboo* data it was 161 (26.83%) invalid paraphrases; in most cases, the “invalidity” was caused by a semantic mismatch between the paraphrase and the seed.
> - Falcon-40B *prompt* data had 103 (17.17%) invalid paraphrases and for the *taboo* it was 177 (29.5%) invalid paraphrases; mostly due to the model treating the paraphrasing task like a conversation, answering to the seed sentence rather than paraphrasing it, thus ignoring the instructions
>
> **Overall, Falcon-collected outcomes did not match the quality of ChatGPT’s.** Compared to ChatGPT, where no invalid paraphrases were detected during our manual evaluation, the Falcon-collected outputs require a lot more manual validation and filtering.
>
> In the paper itself, we plan to use the additional page to highlight this comparison of open source LLMs vs. ChatGPT for this task with further explanation of methodology and discussion of additional results after our manual validation is done. The additional results will entail: lexical diversity, syntactical diversity, no. valid samples and no. samples with wrong paraphrases per each Falcon model variation.
>
> > The influence of parameters of using ChatGPT for this task is not profoundly investigated.
>
> > Question A: How do you design your parameters for using ChatGPT API?
>
> We have investigated several existing works to design the parameter setup. We also kept default values, where related work didn’t advise otherwise. The outcomes of the experiments did not required alterations of the parameter setups w.r.t. research questions.
>
> >Specifically for example, why do you choose the temperature to 1 which means the output is random?
>
> The Temperature parameter is mostly referred to as the one controlling ‘randomness’, ‘diversity’ and ‘creativity’ of the output.  In our research, we wanted to maximize diversity and therefore went with the highest Temperature setting. As the resulting paraphrases were valid and were adhering to the seed sentences and intents, we did not consider other Temperature settings.
>
> >Question B: What is the arrow(↑)'s meaning which is shown after TED in Table 1?
>
> The meaning of the symbol is that the higher values are considered better. We will add an explanation to the table caption.
>
> >The writing of this paper is a bit colloquial. The tense of some sentences is inconsistent. For example, line 281, "We also observed that ChatGPT ...": Past Tense line 333, "We also compare ...": Present Tense
>
> For the camera ready version, we will do a thorough check on the language, including tense harmonization.

---

### Official Review · Reviewer_YaLe · 2023-08-08

**Typos Grammar Style And Presentation Improvements:** N/A
**Soundness:** 4

**Excitement:**

4: Strong: This paper deepens the understanding of some phenomenon or lowers the barriers to an existing research direction.

**Missing References:**

N/A

**Paper Topic And Main Contributions:**

*ChatGPT to Replace Crowdsourcing of Paraphrases for Intent Classification: Higher Diversity and Comparable Model Robustness* presents a refresh to a previous study Larson et al. for intent classification using a relatively nascent technology: ChatGPT. The purpose of this study primarily investigates three research questions. Namely, (1) Can ChatGPT generate valid solutions to a paraphrasing task, given similar instructions as crowd workers (from a previous study)? (2) How do ChatGPT and crowd solutions compare in terms of their lexical and syntactical diversity? (3) How do ChatGPT and crowd solutions compare in terms of robustness of intent classification models trained on such data. The paper is motivated by the prevalence of ChatGPT and akin LLMs while noting the expenses and quality concerns of crowdsourcing. This work follows an apples to apples comparison as best as the authors could to compare with Larson 2020. The paper finds that, yes, ChatGPT is a suitable alternative at a fraction of the price for this particular study, and it generates more diverse summaries than the human annotators. The model quality improves on downstream training for intent classification in most datasets in a significant manner, demonstrating that ChatGPT in this study is a suitable replacement on quality alone. Factoring in the cost comparison, the authors make a convincing argument about the replacement of crowdsourced efforts with LLMs.

**Questions For The Authors:**

Q1: In 4.1 last paragraph of page 5, it is discussed that the possible overlaps are checked. Is this the overlap between GPT, human, and original data for cross-set bias between train and test? This is unclear.

Q2: A concern with TED calculations. I appreciate the normalization between intents, however if the intents can be vague, which would make the normalization less appropriate. To elaborate, how detailed are the intents or specific acts? The examples in Table 2 seem to be somewhat specific, which is good for any takeaways regarding TED. So, if intent "phone" refers to many entities or actions such as texting, calling, using apps, the device itself, then the TED values lumped together would not really tell us how diverse the responses are, simply that there are many diverse actions that are lumped into an intent.

Q3: How did you manual validate the ChatGPT paraphrases? It is hard to believe that over 10,000 statements were manually validated. Just some details suffice.

**Reasons To Accept:**

- The paper examines previously studied crowdsource work under a new lens. This work brings new insights to crowdsource efforts and the roles LLMs may play in synthetic data generation.

- The experiments regarding lexical and syntactic diversity are interesting. These experiments give insights to the quality of data one can likely see as LLMs are used in place of human crowdsource workers.

- The experiments in OOD appear sound and encouraging. The synthetic data brings measurable improvements in OOD datasets for intent classification in BERT.

- The experiments list confidence internal which shows a significant improvement with little to none overlap in CI.

- Clear experiments with a quality discussion of results

- Limitations of work are thoughtfully discussed.

- The paper contributes data and code that can be used in future works.

**Reasons To Reject:**

- The generalization of model robustness findings *should* be tested in other models and architectures. Given the code provided, future work is likely doable in intent classification and other applications.



**Reproducibility:**

4: Could mostly reproduce the results, but there may be some variation because of sample variance or minor variations in their interpretation of the protocol or method.

**Reviewer Confidence:**

4: Quite sure. I tried to check the important points carefully. It's unlikely, though conceivable, that I missed something that should affect my ratings.

---

> ### Author Rebuttal · Authors · 2023-08-28
>
> Thank you very much for your assessment. Below, we address each question or critical remark individually.
>
> > Q1: In 4.1 last paragraph of page 5, it is discussed that the possible overlaps are checked. Is this the overlap between GPT, human, and original data for cross-set bias between train and test? This is unclear.
>
> These overlaps have been computed on the entirety of the GPT, human and original data, without distinction between train and test data. We understand that this is not mentioned in the text and we will fix this in the camera ready version.
>
> > Q2: A concern with TED calculations. I appreciate the normalization between intents, however if the intents can be vague, which would make the normalization less appropriate. To elaborate, how detailed are the intents or specific acts? The examples in Table 2 seem to be somewhat specific, which is good for any takeaways regarding TED. So, if intent "phone" refers to many entities or actions such as texting, calling, using apps, the device itself, then the TED values lumped together would not really tell us how diverse the responses are, simply that there are many diverse actions that are lumped into an intent.
>
> We will clarify the TED calculations (also  will be explained more clearly in the paper for the camera ready version): TED is first calculated pair-wise for each phrase in the intent and data split (separate for human, GPT, original). This is then averaged to get the mean – this should represent how syntactically diverse the phrases are - the higher the number of mean TED is, the more diversity is present.
>
> The issue of having “too general” intents in the data could potentially have the same effect on the lexical diversity – it would only tell us that we have diverse actions lumped under one intent. This is an issue of the existing datasets, where some intents can be too general, while others are overly specific. We, however, have not noted any such overly general intents in the 5 datasets which we used, so the computations of TED and lexical diversity should not be affected by the phenomenon mentioned by the reviewer. E.g. The intent ‘phone’ simply refers to one action, namely calling the banking institution in the financial dataset and not other phone communication actions.
>
> We will clarify the intents used in our experiments by adding a Table of used intents for each dataset in the Appendix for the camera ready version together with examples and clarifications about each intent.
>
> > Q3: How did you manual validate the ChatGPT paraphrases? It is hard to believe that over 10,000 statements were manually validated. Just some details suffice.
>
> To validate the created paraphrases we used a simple web application that we developed for this task. The user, who was one of the authors, was shown the seed samples, from which ChatGPT generated the paraphrases, and one particular paraphrase to validate. The author was then able to either mark the paraphrase as valid or not, with an additional optional checkbox to label the paraphrase as ‘borderline case’ for possible revisions. As the seed sentence changed only once in a while (one time for each 5-20 paraphrases) this significantly reduced the cognitive load on the annotator. Even with such reduction, the entire validation process took a few days (less than a week), as we also discussed the ‘borderline cases’ where the user was not sure about the validity of created paraphrases.
>
> In the camera ready version, we will add details about this process into the paper.
>
> > The generalization of model robustness findings should be tested in other models and architectures. Given the code provided, future work is likely doable in intent classification and other applications.
>
> Experimenting with more generators and classifiers is a limitation that we hope to address in the future work. Fortunately, it didn’t prevent us from answering our research questions, which aimed at comparing the performance of the crowd vs. some widely known and used LLM to investigate if such LLMs could pose a threat to crowdsourcing by producing high quality (valid and diverse) paraphrases. The performance of ChatGPT was, in our opinion, so convincing that we did not need to compare the crowd to other LLMs. However, investigation of paraphrasing capabilities of other LLMs is a logical step to take and a future work for us.
>
> Prompted by the review(s), **we conducted new experiments with both Falcon-7B-instruct and Falcon-40B-instruct** (highest ranking models on the Huggingface benchmark at the time of submission of the paper). We will incorporate the results into the final version of the paper.
>
> We used the same setup as in the diversity experiments - replicating the method of Larson et al. 2020 for collecting paraphrases for 10 different financial intents. We did not conduct any specific parameter tuning, used the same parameter values as for ChatGPT and used the same prompts as we did for the ChatGPT experiments.
>
> We preprocessed the data - removed duplicates and for *taboo* split removed paraphrases which contained tabooed words. Next, as we did not have time to fully validate ~5.5k paraphrases, we randomly chose 600 paraphrases for each split per model to be annotated for validity (is this a valid paraphrase given intent and seed sentence?). The 4 annotated datasets were Falcon-7B prompt and taboo; and Falcon-40B *prompt* and *taboo*. The randomly chosen 600 paraphrases per dataset were annotated by one of the authors.
>
> Summary of our findings:
>
> - Both Falcon models struggled with producing a unified format for the paraphrases, resulting in a some additional overhead in parsing the results;
> - Both Falcon models struggled to produce paraphrases and follow the basic instructions at times.
> - For the Falcon-7B collected *prompt* data, 92 out of the 600 (15.33%) sampled paraphrases were invalid and for the *taboo* data it was 161 (26.83%) invalid paraphrases; in most cases, the “invalidity” was caused by a semantic mismatch between the paraphrase and the seed.
> - Falcon-40B *prompt* data had 103 (17.17%) invalid paraphrases and for the *taboo* it was 177 (29.5%) invalid paraphrases; mostly due to the model treating the paraphrasing task like a conversation, answering to the seed sentence rather than paraphrasing it, thus ignoring the instructions.
>
> **Overall, Falcon-collected outcomes did not match the quality of ChatGPT’s.** Compared to ChatGPT, where no invalid paraphrases were detected during our manual evaluation, the Falcon-collected outputs require a lot more manual validation and filtering.
>
> In the paper itself, we plan to use the additional page to highlight this comparison of open source LLMs vs. ChatGPT for this task with further explanation of methodology and discussion of additional results after our manual validation is done. The additional results will entail: lexical diversity, syntactical diversity, no. valid samples and no. samples with wrong paraphrases per each Falcon model variation.

---

### Official Review · Reviewer_HLjE · 2023-08-08

**Soundness:** 3

**Excitement:**

4: Strong: This paper deepens the understanding of some phenomenon or lowers the barriers to an existing research direction.

**Paper Topic And Main Contributions:**

This paper assesses the ability to use ChatGPT to replace crowd workers in paraphrase generation, specifically for training intent classification models. They follow a 3-round data collection methodology following prior works using taboo words to encourage diversity, and find that ChatGPT generates more diverse data (both lexically and syntactically). The authors train BERT and SVM models using the generated paraphrases and look at the accuracy on OOD data, and find that they achieve comparable/better results on most datasets compared to models trained on the original human data.

**Questions For The Authors:**

Is there a reason you chose to focus on paraphrase generation specifically?

How does using ChatGPT compare with paraphrases generated using other LLMs such as GPT-3 and open-source models like Falcon and Vicuna?

Have you tried other diversity-encouraging methods other than taboo words?

Have you tried using ChatGPT itself for intent classification? Is there really a point in generating additional paraphrases for it if the model itself may be strong enough already at the end task?

**Reasons To Accept:**

The paper is a valuable contribution as it helps shed light on an important question: whether ChatGPT can be used to replace crowdsourcing, particularly for obtaining paraphrases. This is important going forward, as ChatGPT will likely be used more and more as a source for acquiring data.

The data collection and experiments are well planned and executed, following prior methodology (e.g. 3-round approach) for data collection. There is a decently extensive analysis of the generated paraphrases in terms of diversity, validity, and other characteristics.

Lastly, there is a detailed and comprehensive limitations section, showing that the authors have reflected heavily on the limitations and potential future improvements of their work.

**Reasons To Reject:**

The authors only focus on paraphrase generation. It would have been nice to see a bit broader set of use cases or tasks.

Further, they do not compare the generated paraphrases to other LLMs such as GPT-3 and open-source models like Falcon and Vicuna, which would shed additional insights into 1) how much better ChatGPT and GPT-4 are than these other LLMs, 2) how much is it worth paying the OpenAI API costs compared to using open-source models.

Also, it would have been nice to see the authors try training models such as autoregressive LLMs and not only BERT-large and SVM.

It would have been nice to see the authors attempt to compute their own taboo words based on ChatGPT paraphrases from previous rounds, rather than using the taboo words from existing work. On a related note, it would have been nice to see other diversity-encouraging methods other than just taboo words.

Lastly, there is a bit of circular logic here. I have a strong suspicion that ChatGPT may be strong enough at intent classification itself. Hence, rather than generating additional paraphrases to train other intent classification models, using ChatGPT may be enough. I understand the point of this paper is to test the ability of ChatGPT to generate additional paraphrases to train other models for tasks such as intent classification, but when ChatGPT itself is likely good enough for most end tasks, I question the value of this - especially the way it's framed in this paper, which is to use the generated paraphrases to train intent classification models.

**Reproducibility:**

4: Could mostly reproduce the results, but there may be some variation because of sample variance or minor variations in their interpretation of the protocol or method.

**Reviewer Confidence:**

3: Pretty sure, but there's a chance I missed something. Although I have a good feel for this area in general, I did not carefully check the paper's details, e.g., the math, experimental design, or novelty.

---

> ### Author Rebuttal · Authors · 2023-08-28
>
> Thank you very much for your assessment. Below, we address each question (or critical remark) individually.
>
> > The authors only focus on paraphrase generation. It would have been nice to see a bit broader set of use cases or tasks.
>
> > Is there a reason you chose to focus on paraphrase generation specifically?
>
> In our research, we are interested in dataset augmentation (for increasing model robustness through better data). Specifically, we are interested in creation of adversarial and out-of-distribution samples used for training. In this scope, paraphrasing is one of the most used methods. We were originally investigating human computation (crowdsourcing) and human-in-the-loop methods. The advent of ChatGPT and LLMs prompted us to investigate their capabilities in paraphrasing. However, we didn’t build the paper's motivation following this line. Instead, we felt that investigating the impact of LLMs on crowdsourcing is more straightforward (and more important).
>
> Of course, the capabilities of ChatGPT to solve other typical crowdsourcing tasks should be investigated, but this remains a future work as it would be a bit out of scope of this paper. We deem it as a fruitful challenge with lots of research opportunities: for every crowdsourcing task, one needs a corresponding set of reference human-originating data. Some suitable crowdsourcing studies (like of Larson et al) addressing other tasks probably could be identified, however, many more would require additional crowdsourcing in order to systematically map the field. We hope that more studies like ours will be commenced in the future.
>
> > Further, they do not compare the generated paraphrases to other LLMs such as GPT-3 and open-source models like Falcon and Vicuna, which would shed additional insights into 1) how much better ChatGPT and GPT-4 are than these other LLMs, 2) how much is it worth paying the OpenAI API costs compared to using open-source models.
>
> > How does using ChatGPT compare with paraphrases generated using other LLMs such as GPT-3 and open-source models like Falcon and Vicuna?’
>
> The focus of our current paper was to compare the crowd vs. some widely known and used LLM to investigate if such LLMs could pose a threat to crowdsourcing by producing high quality (valid and diverse) paraphrases. We have therefore selected the most popular and accessible LLM at the time - ChatGPT. As its performance outperformed humans in our experiments, it sufficiently answered our research questions (without the need of investigating further LLMs). However, investigation of paraphrasing capabilities of other LLMs is a logical step to take and a future work for us.
>
> Prompted by the review(s), **we conducted new experiments with both Falcon-7B-instruct and Falcon-40B-instruct** (highest ranking models on the Huggingface benchmark at the time of submission of the paper). We will incorporate the results into the final version of the paper.
>
> We used the same setup as in the diversity experiments - replicating the method of Larson et al. 2020 for collecting paraphrases for 10 different financial intents. We did not conduct any specific parameter tuning, used the same parameter values as for ChatGPT and used the same prompts as we did for the ChatGPT experiments.
>
> We preprocessed the data - removed duplicates and for *taboo* split removed paraphrases which contained tabooed words. Next, as we did not have time to fully validate ~5.5k paraphrases, we randomly chose 600 paraphrases for each split per model to be annotated for validity (is this a valid paraphrase given intent and seed sentence?). The 4 annotated datasets were Falcon-7B prompt and taboo; and Falcon-40B *prompt* and *taboo*. The randomly chosen 600 paraphrases per dataset were annotated by one of the authors.
>
> Summary of our findings:
>
> - Both Falcon models struggled with producing a unified format for the paraphrases, resulting in a some additional overhead in parsing the results;
> - Both Falcon models struggled to produce paraphrases and follow the basic instructions at times.
> - For the Falcon-7B collected *prompt* data, 92 out of the 600 (15.33%) sampled paraphrases were invalid and for the *taboo* data it was 161 (26.83%) invalid paraphrases; in most cases, the “invalidity” was caused by a semantic mismatch between the paraphrase and the seed.
> - Falcon-40B *prompt* data had 103 (17.17%) invalid paraphrases and for the *taboo* it was 177 (29.5%) invalid paraphrases; mostly due to the model treating the paraphrasing task like a conversation, answering to the seed sentence rather than paraphrasing it, thus ignoring the instructions
>
> **Overall, Falcon-collected outcomes did not match the quality of ChatGPT’s.** Compared to ChatGPT, where no invalid paraphrases were detected during our manual evaluation, the Falcon-collected outputs require a lot more manual validation and filtering.
>
> In the paper itself, we plan to use the additional page to highlight this comparison of open source LLMs vs. ChatGPT for this task with further explanation of methodology and discussion of additional results after our manual validation is done. The additional results will entail: lexical diversity, syntactical diversity, no. valid samples and no. samples with wrong paraphrases per each Falcon model variation.
>
> > Also, it would have been nice to see the authors try training models such as autoregressive LLMs and not only BERT-large and SVM.
>
> While this was not the focus of our paper, it is an interesting topic for future work. In our paper, we kept the selection of the classifiers the same as in Larson et al. 2020 as we were more interested in the relative comparison of crowd and ChatGPT and less in achieving the best possible classifier performance. However, future work should indeed try to maximize downstream task performance (including experimentation with other classifiers, but also generator LLMs).
>
> > It would have been nice to see the authors attempt to compute their own taboo words based on ChatGPT paraphrases from previous rounds, rather than using the taboo words from existing work.
>
> As suggested, we have considered using taboo words based on ChatGPT paraphrases from previous rounds (either instead or along with the original taboo words). However, in the end we stuck with reusing the ones from Larson et. al to strengthen the comparison. We dismissed using both in order to keep the study simple.
>
> > Have you tried other diversity-encouraging methods other than taboo words?
>
> Similarly, as with the classifier model selection, we have opted for taboo words only because we were replicating the original study and were pursuing a direct comparison between the performance of crowd and ChatGPT. At the same time, taboo words are the most widely used technique for human computation scenarios (per related work). Of course, the exploration of other options is relevant to future work, especially due to the possibility that LLMs may process the diversity cues differently than humans.
>
> > Lastly, there is a bit of circular logic here. I have a strong suspicion that ChatGPT may be strong enough at intent classification itself. Hence, rather than generating additional paraphrases to train other intent classification models, using ChatGPT may be enough. I understand the point of this paper is to test the ability of ChatGPT to generate additional paraphrases to train other models for tasks such as intent classification, but when ChatGPT itself is likely good enough for most end tasks, I question the value of this - especially the way it's framed in this paper, which is to use the generated paraphrases to train intent classification models.
>
> > Have you tried using ChatGPT itself for intent classification? Is there really a point in generating additional paraphrases for it if the model itself may be strong enough already at the end task?
> Investigating the ability of LLMs to perform intent classification is a relevant future work, yet outside of the scope of our current paper. Using an LLM for a downstream task directly makes sense in many cases (it is straightforward and can be done in zero-shot or in-context-learning setup). At the same time, using LLMs as training set generators opens up possibilities to create cheaper (lesser) classifiers and allows more quality control over the data.
>
> That is why the actual downstream task is not our main focus in this paper. Our real concern is the quality of the created paraphrases (as we mention above, our research focus and research questions are motivated by crowdsourcing and its future in the age of accessible LLMs). That is why our experiments are not strictly focused on intent classification: our prompts and prompting techniques have nothing to do with the specific task of intent classification and can be used for any paraphrasing (similar to the original study, from which we reuse the prompts). Of course, it might be the case that ChatGPT can create such good paraphrases because it can implicitly detect the intent.

---

### Meta-Review · Area_Chair_1QC6 · 2023-09-17

**Recommendation:** 4

**Metareview:**

The paper in question aims to evaluate the ability of ChatGPT to replace crowdsourced human workers in generating paraphrases for training intent classification models.

Pros:
1. All the reviewers acknowledge that the paper addresses a relevant and important topic concerning the potential of ChatGPT as an alternative to traditional crowdsourcing methods for obtaining paraphrases.
2. The study adopts a data collection methodology consistent with previous studies, providing a basis for comparison. The detailed analysis of generated paraphrases on parameters like diversity and validity adds credibility to the findings.
3. There is a thorough examination of the limitations of the study, suggesting the authors' depth of understanding and potential areas of improvement.

Cons:
1. Limited Scope: Reviewers also point out that the focus remains solely on paraphrase generation. A broader range of tasks would have offered more comprehensive insights.
2. The omission of a comparison between ChatGPT and other open-source LLMs, like Falcon, Vicuna, is required by reviewers. I can see that the authors add new experiments with Falcon-7B-instruct and Falcon-40B-instruct during rebuttal. These new results should be added to the camera-ready version.

---

### Decision · Program_Chairs · 2023-10-07

**Decision:**

Accept-Main

**Comment:**

The paper in question aims to evaluate the ability of ChatGPT to replace crowdsourced human workers in generating paraphrases for training intent classification models.

Pros:
1. All the reviewers acknowledge that the paper addresses a relevant and important topic concerning the potential of ChatGPT as an alternative to traditional crowdsourcing methods for obtaining paraphrases.
2. The study adopts a data collection methodology consistent with previous studies, providing a basis for comparison. The detailed analysis of generated paraphrases on parameters like diversity and validity adds credibility to the findings.
3. There is a thorough examination of the limitations of the study, suggesting the authors' depth of understanding and potential areas of improvement.

Cons:
1. Limited Scope: Reviewers also point out that the focus remains solely on paraphrase generation. A broader range of tasks would have offered more comprehensive insights.
2. The omission of a comparison between ChatGPT and other open-source LLMs, like Falcon, Vicuna, is required by reviewers. I can see that the authors add new experiments with Falcon-7B-instruct and Falcon-40B-instruct during rebuttal. These new results should be added to the camera-ready version.